# Simple Prediction of Metabolic Equivalents of Daily Activities Using Heart Rate Monitor without Calibration of Individuals

**DOI:** 10.3390/ijerph17010216

**Published:** 2019-12-27

**Authors:** Yuko Caballero, Takafumi J. Ando, Satoshi Nakae, Chiyoko Usui, Tomoko Aoyama, Motofumi Nakanishi, Sho Nagayoshi, Yoko Fujiwara, Shigeho Tanaka

**Affiliations:** 1Department of Food and Nutritional Science, Ochanomizu University, Tokyo 112-8610, Japan; fujiwara.yoko@ocha.ac.jp; 2Department of Nutrition and Metabolism, National Institute of Health and Nutrition, National Institutes of Biomedical Innovation, Health and Nutrition, Tokyo 162-8636, Japan; takafumi.andu@gmail.com (T.J.A.); tomokom@nibiohn.go.jp (T.A.); 3Graduate School of Engineering Science, Osaka University, Osaka 560-8531, Japan; snakae@bpe.es.osaka-u.ac.jp; 4Faculty of Sport Sciences, Waseda University, Saitama 359-1192, Japan; c.usui@aoni.waseda.jp; 5Omron Healthcare Co., Ltd., Kyoto 617-0002, Japan; motofumi.nakanishi@omron.com (M.N.); sho.nagayoshi@omron.com (S.N.)

**Keywords:** physical activity intensity, physical fitness, %heart rate reserve, resting heart rate, leave-one-out method

## Abstract

Background: Heart rate (HR) during physical activity is strongly affected by the level of physical fitness. Therefore, to assess the effects of fitness, we developed predictive equations to estimate the metabolic equivalent (MET) of daily activities, which includes low intensity activities, by % HR reserve (%HRR), resting HR, and multiple physical characteristics. Methods: Forty volunteers between the ages of 21 and 55 performed 20 types of daily activities while recording HR and sampling expired gas to evaluate METs values. Multiple regression analysis was performed to develop prediction models of METs with seven potential predictors, such as %HRR, resting HR, and sex. The contributing parameters were selected based on the brute force method. Additionally, leave-one-out method was performed to validate the prediction models. Results: %HRR, resting HR, sex, and height were selected as the independent variables. %HRR showed the highest contribution in the model, while the other variables exhibited small variances. METs were estimated within a 17.3% difference for each activity, with large differences in document arrangement while sitting (+17%), ascending stairs (−8%), and descending stairs (+8%). Conclusions: The results showed that %HRR is a strong predictor for estimating the METs of daily activities. Resting HR and other variables were mild contributors. (201 words)

## 1. Introduction

Obesity and lifestyle diseases are worldwide problems [1]. Physical inactivity and/or low fitness levels have an adverse effect on one’s health status, and contribute to chronic diseases such as diabetes and cardiovascular diseases [2]. In modern society, daily activities such as commuting and housework account for the majority of total energy expenditure (EE), far more than exercise [3,4]. Therefore, an accurate measure of daily activities, especially its duration and metabolic intensity, would be crucial both in assessing individual activity levels and in evaluating the independent effects of the daily activities on health status in an epidemiological study.

Recently, accelerometers and other devices with a variety of algorithms for estimating EE have been expanded in the market. However, tri-accelerometers and other devices are still expensive, and an accelerometer cannot measure specific activities, like ascending the stairs and carrying packages, at present [5]. On the other hand, a heart rate (HR) monitor is a reasonable device and many products with HR monitors are being developed. HR during physical activity has a strong correlation with the level of physical fitness [6,7]. Therefore, HR monitors can predict EE during exercise if the prediction equation is established for each individual. However, this method utilizing the prediction equation is not useful for a large-scale study because it has to be calibrated by measuring EE with HR for each individual. Furthermore, the accuracy of the model is decreased for low levels of activity [8], and HR differs depending on age, sex, and the level of physical fitness [9]. Therefore, these characteristics are key obstacles that prevent us from accurately estimating EE. To overcome this shortcoming, Keytel et al. developed an equation using multiple-regression analyses, which included age, weight, and sex as independent variables to estimate EE without the need of calibration for each individual [10]. Charlot et al. later improved Keytel’s equation and established a more accurate assessment by adding the measured maximal HR (HRmax) and real running speed [11]. However, these models can only be applied for exercises such as walking and running, and cannot be applied to daily activities. Furthermore, HR in low activities under 95 bpm cannot be used to predict EE accurately [12], and it is easily affected by mental stress especially in this range [13]. However, some variables related to HR may contribute to improve the prediction of EE by being added to the multiple regression equation.

Percent HR reserve (%HRR) has been reported to be a major predictor for estimating EE. It is a relative value, assuming that HRmax is 100% and resting HR is 0%; thus, it reflects one’s exercise intensity, minimizing individual differences. However, most studies using %HRR were conducted during specific activities such as walking, slope walking using a treadmill, and jogging. Few reports have assessed the contribution of %HRR to EE regarding daily activities such as vacuuming, ascending stairs, and walking with a load [14,15]. Therefore, assessing the relationship between %HRR and EE in daily activities is critical. Additionally, since the slope of %HRR would be influenced by age and fitness levels [16], these variables should be considered. Resting HR is known to reflect one’s level of fitness and age [17]. Thus, predictions accounting for those two parameters may have the potential to improve the accuracy of the equations. However, %HRR and resting HR may have a slight correlation. Therefore, statistical analysis should be conducted carefully while accounting for multicollinearity.

EE is an absolute scale expressed in the unit of kcal or kJ per minute as physically induced energy expenditure. On the other hand, metabolic equivalent (MET) is an index of physical activity intensity, calculated as a ratio of metabolic rate during an activity to metabolic rate at rest, and serves as a normalized index of physical activity intensity in each individual. Therefore, it would be convenient if METs could be calculated directly from measured values, without the need for recalculation utilizing EE. Furthermore, it would be feasible to apply MET to the recommendation of daily physical activities [18].

The aim of this study was to develop simple multiple-regression models for estimating METs of daily activities by including parameters such as %HRR and resting HR in adults. We also validated our model utilizing previous studies by comparing the differences.

## 2. Materials and Methods

### 2.1. Participants

Forty-two volunteers participated in this study, which was conducted at the National Institute of Health and Nutrition in Japan. They were physically healthy to complete the tests, without having any movement disorders and suffering from any cardiac disorders. Four to six male and female volunteers were recruited from each of the four 10-year age groups (20–29, 30–39, 40–49, 50–59). The protocol was approved by the Ethical Committee of the National Institute of Health and Nutrition and written informed consent was obtained from the participants (No. 20140226-01).

### 2.2. Protocol

The procedure for measuring HR and exhalation gas was as follows: participants came to the institution in the morning following an overnight fast. Height was measured to the nearest 0.1 cm without shoes and socks. Body weight was measured by a weight scale with a digital bioelectrical impedance analyzer (HBF-362, Omron, Kyoto, Japan) with light clothes. Body mass index (BMI) was calculated as body weight (kg) divided by height in meters squared (m^2^). After collecting anthropological measurements, a HR monitor called Health Patch MD Sensor (VitalConnect, San Jose, CA, USA) and a facemask connected to a Douglas bag were placed on the participants. The MD sensor consists of two electrocardiographic (ECG) electrodes, and it detects QRS waves. The R-R intervals are computed from the time interval between QRSs. More details of the protocol are described in our previous study [19]. The sensor was validated by Chan et al. [20]. Resting HR was measured over 7 min while the test subjects were seated. The resting HR and HR during each activity were converted from the recorded R-R interval. Empty data or outliers (±4 SD in radio calisthenics, ±3 SD in other activities) of the epoch (0.4 s) were eliminated [21,22]. Calculated HR was the average of every minute for each activity. The O_2_ and CO_2_ concentrations of expired gas were measured by a mass spectrometer (ARCO-2000, Arco System, Chiba, Japan) and gas volume was measured by a dry gas meter (DC-5, Shinagawa, Tokyo, Japan). The EE was calculated using Weir’s equation and was used as a reference value [23]. Measuring real HRmax is not feasible for elderly and sedentary subjects. Therefore, we predicted HRmax using Tanaka’s equation [24].

The resting metabolic rate (=1 MET) was measured at least twice while the subjects were seated for each 7-min interval to check for stability while measuring resting HR. The stability state was determined when the values of respiratory gas were comparable both times. Each value of METs was calculated by dividing by the EE value recorded at rest. Table 1 shows measurement parameters and their methods. All participants performed 20 daily activities as shown in Table 2. At least four 5-min breaks were set during the test to prevent the influence of previous activities. Participants were allowed to drink bottled water, which was provided during the breaks.

### 2.3. Model Development

Before developing statistical models, we calculated correlation coefficients between HR and METs, and between %HRR and METs for all subjects, and compared the two *r* values. This was necessary because there is a strong correlation between HR and %HRR (0.931), and only one of them should be used in the model. The *r* value with METs for %HRR was 0.938 and higher than that of HR (0.854). Thus, we chose %HRR as the variable for developing models. We also added six potential parameters based on the evidence obtained so far [9,10,11,25,26]. Candidate independent variables and the process of choosing them are shown in Figure 1. The correlation coefficients between METs and each variable are also shown in Figure 1. We developed two types of models, one with BMI and the other with weight and height, because BMI has a strong correlation with weight (*r* = 0.887) and height (*r* = 0.241). We computed all possible combinations of variables using the brute force method. After that, we applied Akaike’s information criterion (AIC); the model with the minimum value of AIC gives the best fit of all models, deleting redundant variables [27]. The models selected by AIC also showed the highest adjusted R^2^ in our case. The criteria for checking multi-collinearity was that the sign of the coefficients of variables did not change from the correlation coefficient of the dependent variable. We also calculated the variance inflation factor (VIF) of each model and ensured that those values were all less than 1.1. Thus, there was no possibility of having multi-collinearities in the models [28].

### 2.4. Validation Test and Statistical Analysis

The leave-one-out method was performed to validate the prediction models. In the present study, we developed models using all subjects except one subject, and tested for validation by applying this left out subject, and repeated it as many times as the number of subjects. Comparisons between measured METs (indirect calorimetry) and estimated METs from the model were performed by Wilcoxon signed rank test. Mean percent error (MPE) and root mean square error (RMSE) were also calculated for each activity by averaging the errors for each leave-out run as below:MPE (%) = (Estimated METs − Measured METs)/Measured METs × 100(1)
(2)$$\mathrm{RMSE}=\sqrt{\frac{1}{n}\overset{n}{\underset{i=1}{\sum }}{\left({\hat{y}}_{i}-y_{i}\right)}^{2}}$$
where:$${\hat{y}}_{i}=\;estimated\;METs$$
$$y_{i\;}=\;measured\;METs$$

The modified Bland and Altman plots were used to depict the agreement between measured METs and estimated METs by leave-one-out [29]. We validated our models by comparing them with other previous studies. The methods for developing a model and for calculating the error of each model were different. Thus, we recalculated RMSE using the hold-out method. We divided all subjects into two parts in a ratio of two to one for the model development group and the validation group, and calculated RMSE of each activity. This was repeated 10,000 times, and the average values of RMSE of each activity among the tests were used for comparison with previous studies. The statistical analyses were performed using SPSS version 25 for Windows (IBM, Armonk, NY, USA) and Excel add-in software “Multi Tahenryo” (Istat, Tokyo, Japan).

## 3. Results

### 3.1. Subject Characteristics

The characteristics of the participants are shown in Table 3. The average height and weight were almost equivalent to those of the Japanese general population. Two participants were excluded because they could not conduct any activities. Some blank HR values appeared in each activity due to measurement failures.

### 3.2. Coefficients of Independent Variables

Table 4 and Table 5 show the results of multiple regression analyses with HR and %HRR, respectively. When comparing models with one variable, HR or %HRR, the *r* value of the model with %HRR was 0.938 and showed a higher value than the one with HR (0.852). In Table 5, %HRR was the highest contributor to the model (standardized *β* = 0.944) among all variables. The standardized *β* of resting HR in the model with %HRR was −0.078 and it showed a smaller contribution to the model. The correlation coefficients (*r*) of the model with %HRR, resting HR, and height had the highest values, which were close to 1 in all models, at 0.943.

Table 6 and Table 7 show the mean percent error (MPE) and root mean square error (RMSE) between measured METs and estimated METs. Figure 2 is a graphical depiction of MPE.

The total MPE of all activities was 2.8% (SD 22.3%) in the model including only %HRR and 2.4% (SD 23.1%) in the model including %HRR, resting HR, and height. METs were estimated within 17.3% of average differences in all activities. METs of the exercise activities of walking and jogging were estimated within ±7% of average differences, and they were estimated more accurately than non-exercise activities. The largest difference in MPE was observed during document arrangement while sitting (16.6–17.3%, *p* < 0.01), and its MET value was overestimated. Ascending and descending stairs also showed large differences of approximately −8% (*p* < 0.01) and 8% (no significant difference), respectively. When comparing measured METs and estimated METs in all activities, there were no significant differences in all models. The model with %HRR, resting HR, and height showed an approximate 2% decrease in MPE during low intensity activities such as operating a mobile phone and PC work, compared to the model with only %HRR.

In contrast to the results of MPE, the values of RMSE increased with the intensity of activities. Ascending stairs and jogging showed high values from 1.14 to 1.67. On the other hand, other activities showed low values of less than 0.76.

Figure 3 depicts the results of modified Bland–Altman analyses between measured and estimated METs. There was no difference between the two models. Jogging of a participant was largely overestimated by 3 METs, while jogging of another participant was underestimated by 4 METs. The confidence interval was ±1.3 in both models and *r* of the model with %HRR, resting HR, and height (B) was −0.32 (*p* < 0.001).

## 4. Discussion

In this study, we developed models for the brief prediction of METs utilizing a new combination of predictors. We predicted METs directly and selected three independent variables, which were %HRR, resting HR, and sex or height. These values of METs can be recalculated to determine EE. Therefore, we assume that the %error of METs and *r* coefficient of METs are similar to those of EE, which have been demonstrated by other studies predicting EE.

Charlot et al. used five predictors to predict EE, including HR, weight, resting HR, real HRmax, and maximal oxygen uptake, and the *r* coefficient of their study was 0.94 [11]. Similarly, the *r* value of our model was 0.943. The reason for this similarity might be because METs is divided by EE at rest while seated, and that could have attenuated individual differences.

The results of our study showed that %HRR was the strongest predictor of a variety of daily activities from low to high intensity, and that resting HR was not a major contributor, though the MPE of low activities slightly improved by 2%. Therefore, the model with only %HRR was enough to estimate METs with relatively small prediction errors. Hiilloskorpi et al. also showed that %HRR was the highest contributor compared to HR or HRnet (HR activity–resting HR) [14]. The results of the present study assured that %HRR is a principle factor for estimating METs during daily activities. From a physiological aspect, METs is obtained by dividing by the respiratory gas values during the sitting state, and that reflects individual resting metabolic rate. Similarly, %HRR is divided by (HRmax–resting HR), which reflects individual levels of fitness. Therefore, it seems suitable to use %HRR for estimating METs. From a mathematical aspect, %HRR contains predicted HRmax and resting HR. The predicted HRmax was calculated utilizing the regression equation based on age, and it accounts for the biggest value of the numerator in the fraction of %HRR. On the other hand, resting HR in the fraction of %HRR was subtracted from both the numerator and denominator. This mathematical variation might explain the high contribution of METs.

Resting HR was hypothesized to reflect individual levels of physical fitness. Only one study investigated resting HR as an independent variable in a multiple regression equation [11]. However, the degree of the contribution of resting HR in the study was not clear. Therefore, in the present study, we calculated standardized *β* to assess the degree of contribution of each variable. In our results, standardized *β* values of resting HR were the second highest among all variables in both models with HR and %HRR, and these *β* values especially demonstrated a high contribution in the model with HR (−0.415). On the other hand, standardized *β* of resting HR in the model with %HRR suggest that resting HR was a minor contributor (−0.074). A possible explanation for this low contribution is that %HRR already contains resting HR in the equation.

Other variables such as sex and height were selected in the present study, as they are known to affect the HR. Sex was selected as a predictor for estimating EE; although more specifically, our estimation was for METs [10,11,26]. As for height, other studies predicting EE showed that weight was a stronger predictor than height because EE is strongly influenced by weight [9,11]. However, in our study, weight was not selected in any of the models in the prediction of METs. Currently, no previous studies that predicted METs using multiple regression analysis included weight as one of their variables. According to the previous study on the influence of body weight, height, age, and sex on total energy expenditure (TEE) [30], TEE, basal metabolic rate (BMR), and activity energy expenditure (AEE) were related to weight and height. However, the influence of weight disappeared when TEE was expressed as physical activity level (PAL, derived as TEE/basal metabolic rate (BMR)), while height and age remained highly significant predictors. As MET has a similar structure to PAL, the influence of weight may disappear as in the case of PAL. Additionally, the characteristics of the participants in the present study were the same as those of the average Japanese population, which means that the percentage of overweight or obese people was low compared to in other countries, and this small deviation might attenuate the influence of weight. Further analysis will be needed in this aspect. Nevertheless, our data showed that standardized *β* values of sex and height were small: 0.049 and 0.065, respectively, and those values were less than one 14th to one 19th that of the standardized *β* of %HRR (0.944).

Figure 4 shows the comparison of prediction errors expressed as RMSE between our study and some previous studies with only a HR algorithm [11,14,31]. RMSEs of previous studies were calculated from measured and estimated METs by substituting values of our study into their original equations. HRmax values by Charlot’s study and %HRR values by Hiilloskorpi’s study were calculated by using their own method. These three prediction methods were selected because they could be regarded as a representative calibration study using HR. We developed one equation that encompassed low to high intensity activities. The equation showed steady accuracy similar to that of Crouter’s in almost all activities, especially in low intensity activities, even though the maximum errors were large. Crouter’s study showed small RMSE values, which may be because they developed two kinds of equations for low and high intensity activities.

Figure 5 depicts estimated METs of validation groups using an equation developed by the hold-out method similar to that of Figure 4, and estimated METs from previous studies. Estimated METs from the present study had a similar trend as measured METs from other studies. However, ascending stairs and jogging were underestimated to the same degree as Crouter’s equation. Additionally, large values of maximum error were observed in ascending stairs and jogging. The other two equations developed by Hiilloskorpi and Charlot overestimated METs for all ranges. In addition to that, Charlot recommended to use his model only for activities with from 25 %HRR to 75 %HRR. It is difficult to clarify the reason for the overestimation, various factors including technical errors can be considered.

Many algorithms using accelerometers and other devices such as combined monitors with an accelerometer, HR monitor, and patch-type sensor module have been developing rapidly [32,33,34,35]. However, those devices cannot measure specific activities like cycling, ascending stairs, and carrying packages. For example, the MPEs of ascending and descending stairs ranged from −61.4% to 40.7% in Ohkawara’s study [35]. In our study, using HR, averaged MPE of ascending/descending was 0.6% in the leave-one-out method. Even if they were separated into two distinct activities, ascending and descending stairs still showed relatively small MPEs (−8% and +8%, respectively). Strath et al. estimated METs by combining an accelerometer and an HR sensor (simultaneous use of HR and motion sensor technique) [36]. However, they did not conduct low intensity activities such as PC work and slow walking. Furthermore, their model only included leg and arm activity, and they used this model for predicting daily activities, which makes it impossible to compare our MPE values to those obtained in their study. However, if we are allowed to compare moderate to vigorous activities in their model to those of the present study, including slow walking, vacuuming, fast walking, and ascending/descending stairs, the MPE values were similar, within 10% difference. Therefore, the fit of our model was as good as a combined model that utilized an accelerometer and a HR monitor. MPEs of our study varied up to 17.3% based on the activity type. The largest MPE in the present study, 17.3%, was observed in participants during document arrangement while sitting. Calculated as EE, it would be 117.0 kcal of overestimation if the average person in this study arranges documents while sitting for eight hours (data not shown in table). It may be an unignorable difference for inactive people. However, our findings showed that the differences were still small compared to other similar studies that used HR and tri-accelerometer, and that our findings were just as accurate as the combined model of HR and accelerometer [10,35,36].

Although MPE was small in total activities, the SD of MPE was large in all models (22.3–23.2%). When comparing SDs for each activity, it ranged from 11.6% to 36.9%. Other studies estimating EE from HR had smaller SDs, from 4.4% to 13.1%; however, the method and predicting unit in those studies were different [10,11]. One possible reason is that our models included a variety of daily activities, compared to models of other studies that were mainly developed from locomotive activities such as walking, jogging, and cycling. Nevertheless, in our study, the 95% prediction interval was from −1.3 to 1.3 METs. This range was as small as the prediction interval obtained by Strath’s study, which used a simultaneous HR-motion sensor technique (within ± 1.5 METs) [36].

We have some limitations in our study. First, each subject performed all 20 activities and the same values of the variables (resting HR, height, and sex) were repeatedly used for the multiple regression models. This might cause an overfitting of the data set, leading to the deformation of models. Second, some activity data, especially jogging data, were missing. Furthermore, there were some outliers for jogging, which might be caused by measurement error. However, almost the same results were obtained even when these outliers were included in the analyses (data not shown in table). Those outliers could slightly contribute to the overestimation or underestimation of METs during jogging activities. To avoid these technical errors, a more accurate HR monitoring system is still needed.

Despite its limitations, the present study showed small RMSEs of METs, at the same level as Crouter’s study using two equations, as well as previous reports that used tri-accelerometer and/or HR. Moreover, ascending and descending stairs showed relatively small MPEs compared to methods using accelerometers. This result might be mainly because METs was predicted directly, instead of EE. Although only %HRR was a large contributor, one should be cautious to conclude that %HRR can simply predict one’s METs accurately, because there are few studies predicting METs that include potential predictors and utilize multiple regression analysis. Overall, this study provides an insight into the degree of contribution of %HRR and other variables for estimating one’s METs. Further studies in free-living conditions are required to accumulate more evidence and assure the accuracy of the models.

## 5. Conclusions

Multiple regression analyses were performed to develop prediction models of METs. %HRR showed the highest contribution in the model. Additionally, resting HR mildly contributed to the present model, contrary to our expectations. When comparing each activity, METs were estimated within a 17.3% difference. The largest average difference was observed for document arrangement while sitting (16.6–17.3% overestimation). The average %difference of METs in total activities was 2.4% (SD 23.1) in the model with %HRR, resting HR, and height. RMSE was also shown to be as small as those of previous studies with HR algorithm. The results showed that METs in daily life can be accurately predicted from %HRR to the same extent as previous reports using tri-accelerometer and/or HR.

## Figures and Tables

**Figure 1 ijerph-17-00216-f001:**
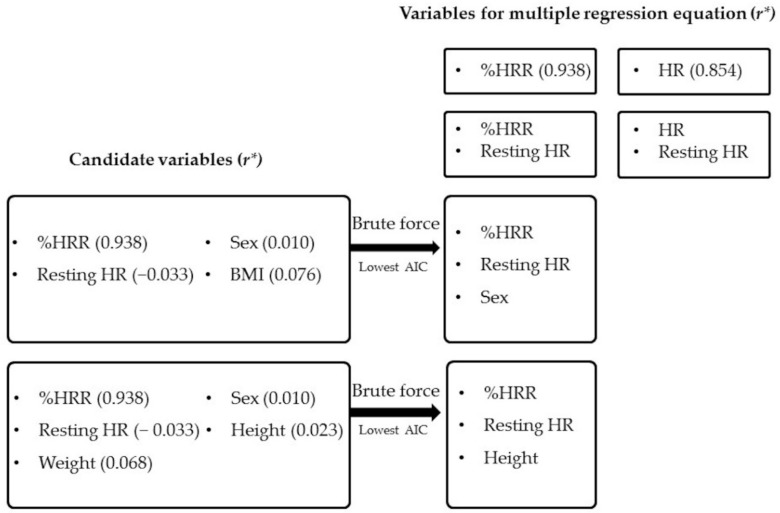
Selection of independent variables (n = 40). Dependent variable was METs. All possible combinations of variables were computed by using the brute force method. After that, the model with the minimum value of Akaike’s Information Criterion (AIC), which gives the best fit of all models, deleting redundant variables was selected. * *r* is the correlation coefficient between METs and each variable.

**Figure 2 ijerph-17-00216-f002:**
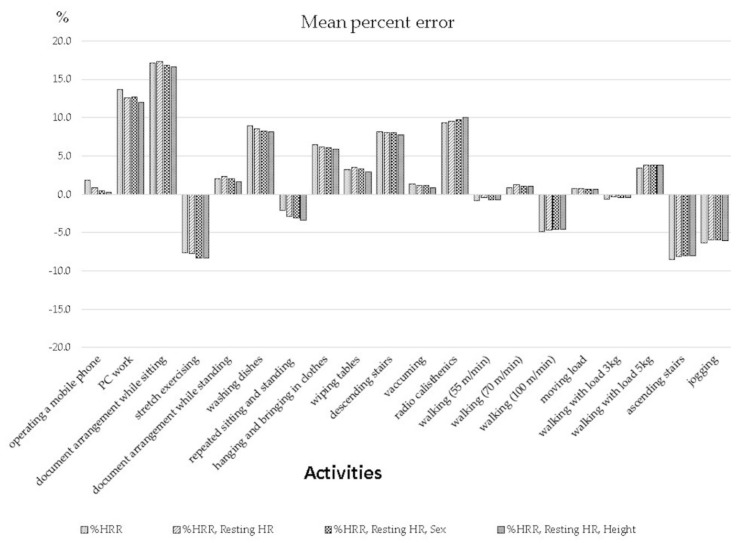
Mean percent error of each equation validated by leave-one-out (n = 40).

**Figure 3 ijerph-17-00216-f003:**
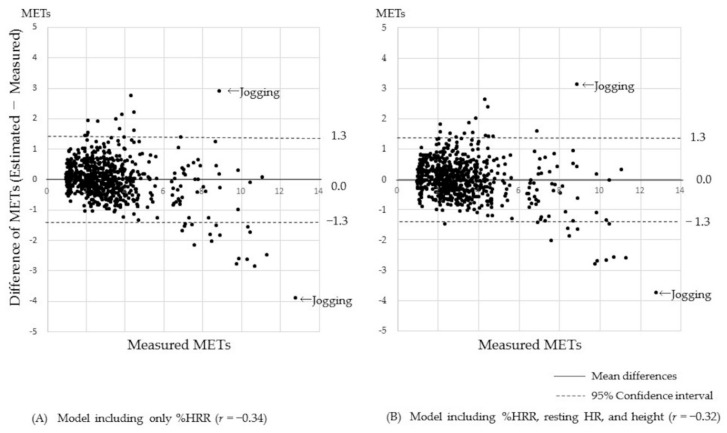
Differences between measured and estimated METs by Bland–Altman analysis (N = 40, 673 dots).

**Figure 4 ijerph-17-00216-f004:**
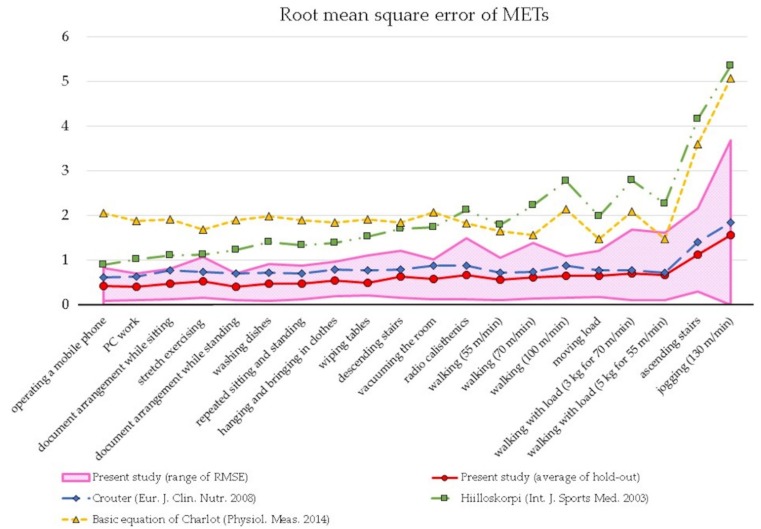
Comparison of prediction errors expressed as RMSE with those of previous studies.

**Figure 5 ijerph-17-00216-f005:**
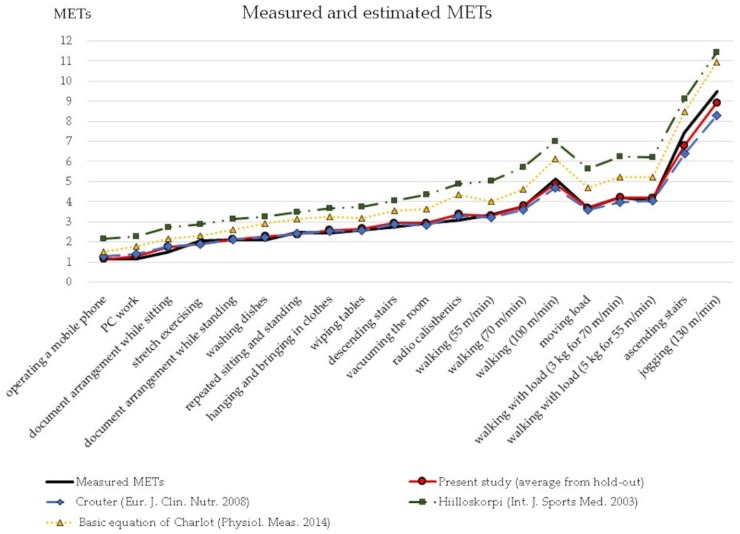
Measured and estimated METs of previous studies and present study.

**Table 1 ijerph-17-00216-t001:** Measurement parameters and measurement methods.

Measurement Parameter	Unit	Method
Resting HR	bpm	Seated state for 7 min and averaged per minute
Predicted HRmax	bpm	208 − 0.7 × age
%HRR	bpm	(HR activity − resting HR)/(predicted HRmax − resting HR) × 100
METs	-	EE of activity/EE of rest in the sitting position

**Table 2 ijerph-17-00216-t002:** Measured METs and %HRR of 20 activities.

Classification	Activities	N	Minutes of Activity (Minutes of Measurement)	METs	%HRR
Exercise	radio calisthenics	25	8 (3.0)	3.1 ± 0.4	21.6 ± 5.3
	walking (55 m/min)	33	5 (2.0)	3.3 ± 0.5	21.8 ± 8.0
	walking (70 m/min)	29	5 (2.0)	3.7 ± 0.5	26.0 ± 8.0
	walking (100 m/min)	30	5 (2.0)	5.1 ± 0.9	36.5 ± 10.8
	jogging (130 m/min)	21	4 (1.0)	9.5 ± 1.5	73.4 ± 12.6
Household work and daily activities	operating a mobile phone	36	7 (5.0)	1.1 ± 0.1	1.1 ± 3.7
	PC work	38	7 (5.0)	1.1 ± 0.1	2.3 ± 3.4
	document arrangement while sitting	39	5 (3.0)	1.5 ± 0.3	6.5 ± 3.4
	stretch exercising	30	12 (5.0)	2.1 ± 0.3	8.0 ± 4.5
	document arrangement while standing	40	5 (3.0)	2.1 ± 0.4	10.2 ± 4.6
	washing dishes	37	5 (3.0)	2.1 ± 0.4	11.7 ± 5.5
	hanging and bringing in clothes	38	5 (2.0)	2.4 ± 0.4	14.5 ± 5.7
	repeated sitting and standing	38	4 (2.0)	2.5 ± 0.3	12.9 ± 5.1
	wiping tables	39	5 (2.0)	2.6 ± 0.5	15.0 ± 5.8
	descending stairs	36	5 (1.5)	2.7 ± 0.4	17.8 ± 6.3
	vacuuming the room	35	3 (2.0)	2.9 ± 0.6	17.7 ± 6.4
	moving load (5 kg bag of rice)	37	5 (2.0)	3.7 ± 0.6	25.4 ± 8.2
	walking with load (5 kg for 55 m/min)	33	5 (2.0)	4.0 ± 0.5	29.7 ± 8.3
	walking with load (3 kg for 70 m/min)	29	5 (2.0)	4.2 ± 0.6	30.1 ± 9.5
	ascending stairs	30	5 (1.0)	7.4 ± 0.9	54.0 ± 7.7

**Table 3 ijerph-17-00216-t003:** Physical characteristics of the participants.

Age Groups (Yr)	N	Age (Years)	Height (cm)	Weight (kg)	BMI (kg/m^2^)	Resting HR in the Sitting Position (bpm)
Male	20	39.5 ± 10.6	171.2 ± 5.4	68.7 ± 12.3	23.3 ± 3.4	67.7 ± 6.6
20–29	6	26.2 ± 3.1	169.0 ± 7.2	66.3 ± 10.6	23.1 ± 2.0	67.0 ± 5.9
30–39	3	37.3 ± 2.1	171.2 ± 3.8	65.1 ±18.6	22.1 ± 5.3	61.4 ± 5.4
40–49	6	43.2 ± 3.9	173.1 ± 6.2	73.1 ± 12.1	24.4 ± 3.6	72.8 ± 6.1
50–59	5	52.2 ± 1.8	171.5 ± 2.3	68.4 ± 13.1	23.2 ± 3.9	67.3 ± 5.9
Female	20	38.0 ± 11.7	159.2 ± 7.1	55.9 ± 12.3	22.0 ± 4.2	67.3 ± 10.3
20–29	5	23.0 ± 2.3	157.3 ± 4.5	49.1 ± 5.1	19.8 ± 1.5	64.0 ± 12.8
30–39	5	33.0 ± 3.5	165.3 ± 10.6	62.2 ± 14.1	22.7 ± 4.3	70.8 ± 5.8
40–49	5	43.0 ± 4.2	155.9 ± 5.7	52.8 ± 17.1	21.6 ± 6.4	66.8 ± 13.7
50–59	5	52.8 ± 1.3	158.1 ± 2.2	59.4 ± 8.0	23.8 ± 3.2	67.6 ± 12.1

Mean ± SD.

**Table 4 ijerph-17-00216-t004:** Results of multiple regression analyses for estimating METs; model with HR (n = 40).

Independent Variables		Intercept	HR	Resting HR	*r*	R*^2^*	SEE *^1^ (MET)
HR	Unstandardized *β* Standard error *p*	−4.030 0.175 <0.001	0.080 0.002 <0.001		0.852	0.725	0.983
	Standardized *β*		0.852				
HR, Resting HR	Unstandardized *β* Standard error *p*	0.679 0.206 0.001	0.095 0.001 <0.001	−0.089 0.003 <0.001	0.934	0.873	0.669
	Standardized *β*		1.009	−0.415			

Dependent variable is METs. *^1^ SEE: standard errors of the estimate.

**Table 5 ijerph-17-00216-t005:** Results of multiple regression analyses for estimating METs; model with %HRR (n = 40).

Independent Variables		Intercept	%HRR	Resting HR	Sex (M = 1, F = 0)	Height	*r*	R*^2^*	SEE *^1^ (MET)
%HRR	Unstandardized *β* Standard error *p*	1.053 0.039 <0.001	0.105 0.001 <0.001				0.938	0.880	0.648
	Standardized *β*		0.938						
%HRR, Resting HR	Unstandardized *β* Standard error *p*	2.123 0.192 <0.001	0.105 0.001 <0.001	−0.016 0.003 <0.001			0.941	0.886	0.634
	Standardized *β*		0.942	−0.074					
%HRR, Resting HR, Sex	Unstandardized *β* Standard error *p*	2.046 0.192 <0.001	0.106 0.001 <0.001	−0.016 0.003 <0.001	0.184 0.048 <0.001		0.942	0.888	0.628
	Standardized *β*		0.944	−0.075	0.049				
%HRR, Resting HR, Height	Unstandardized *β* Standard error *p*	−0.176 0.494 0.721	0.106 0.001 <0.001	−0.017 0.003 <0.001		0.014 0.003 <0.001	0.943	0.890	0.623
	Standardized *β*		0.944	−0.078		0.065			

Dependent variable is METs. *^1^ SEE: standard errors of the estimate.

**Table 6 ijerph-17-00216-t006:** Mean percent error of the predictive equations with leave-one-out method.

Variables in Equation	%HRR	%HRR +Resting HR	%HRR + RestingHR + Sex	%HRR + RestingHR + Height
Activities	MPE (%)	MPE (%)	MPE (%)	MPE (%)
operating a mobile phone	1.9 ± 32.2	0.9 ± 34.5	0.4 ± 36.9	0.2 ± 36.2
PC work	13.7 * ± 30.2	12.6 * ± 32.5	12.7 * ± 33.7	12.0 * ± 33.3
document arrangement while sitting	17.1 ** ± 25.6	17.3 ** ± 28.1	16.9 ** ± 29.8	16.6 ** ± 29.6
stretch exercising	−7.6 ± 23.4	−7.7 ± 24.0	−8.4 ± 24.9	−8.3 ± 24.4
document arrangement while standing	2.0 ± 18.7	2.3 ± 19.7	2.1 ± 20.4	1.6 ± 20.3
washing dishes	8.9 ± 24.2	8.6 ± 22.2	8.3 ± 22.5	8.2 ± 22.4
repeated sitting and standing	−2.1 ± 19.9	−2.9 ± 19.5	−3.1 ± 19.8	−3.4 ± 20.4
hanging and bringing in clothes	6.5 ± 22.4	6.2 ± 21.6	6.1 ± 21.9	5.9 ± 23.2
wiping tables	3.3 ± 19.7	3.5 ± 19.9	3.3 ± 20.5	2.9 ± 19.9
descending stairs	8.2 ± 25.6	8.1 ± 24.8	8.0 ± 24.9	7.8 ± 24.9
vacuuming the room	1.4 ± 22.0	1.1 ± 21.4	1.2 ± 22.8	0.8 ± 22.7
radio calisthenics	9.4 ± 20.7	9.5 ± 20.9	9.8 ± 20.9	10.0 ± 21.0
walking (55 m/min)	−0.8 ± 16.9	−0.4 ± 17.3	−0.7 ± 17.2	−0.7 ± 17.4
walking (70 m/min)	0.9 ± 17.7	1.3 ± 17.6	1.0 ± 17.3	1.0 ± 17.4
walking (100 m/min)	−4.9 ± 12.2	−4.7 ± 12.3	−4.6 ± 11.9	−4.6 ± 11.9
moving load (5 kg bag of rice)	0.8 ± 17.9	0.7 ± 17.2	0.6 ± 17.3	0.7 ± 17.5
walking with load (3 kg for 70 m/min)	−0.6 ± 18.0	−0.3 ± 18.2	−0.5 ± 17.6	−0.4 ± 17.5
walking with load (5 kg for 55 m/min)	3.4 ± 17.0	3.9 ± 17.2	3.8 ± 17.0	3.8 ± 16.7
ascending stairs	−8.5 ** ± 12.3	−8.2 ** ± 11.9	−8.0 ** ± 11.6	−8.0 ** ± 11.7
jogging (130 m/min)	−6.3 ± 15.3	−6.0 ± 15.3	−5.9 ± 15.1	−6.0 ± 14.9
Total activities	2.8 ± 22.3	2.8 ± 22.6	2.6 ± 23.2	2.4 ± 23.1

MPE (%): Mean Percent Error = (Estimated METs − Measured METs)/Measured METs × 100; Mean values were significantly different between measured and estimated METs (Wilcoxon signed rank test): * *p* < 0.05, ** *p* < 0.01.

**Table 7 ijerph-17-00216-t007:** Root mean square error (RMSE) of the predictive equations with leave-one-out method.

Variables in Equation	%HRR	%HRR + Resting HR	%HRR + RestingHR +Sex	%HRR + RestingHR +Height
Activities	RMSE	RMSE	RMSE	RMSE
operating a mobile phone	0.36	0.39	0.42	0.41
PC work	0.38	0.40	0.41	0.40
document arrangement while sitting	0.44	0.46	0.48	0.47
stretch exercising	0.49	0.51	0.53	0.53
document arrangement while standing	0.39	0.40	0.42	0.41
washing dishes	0.52	0.48	0.49	0.49
repeated sitting and standing	0.46	0.45	0.46	0.47
hanging and bringing in clothes	0.54	0.52	0.53	0.55
wiping tables	0.51	0.51	0.52	0.50
descending stairs	0.66	0.64	0.65	0.64
vacuuming the room	0.59	0.56	0.59	0.58
radio calisthenics	0.66	0.66	0.67	0.67
walking (55 m/min)	0.56	0.57	0.57	0.57
walking (70 m/min)	0.64	0.63	0.62	0.63
walking (100 m/min)	0.67	0.66	0.65	0.64
moving load (5 kg bag of rice)	0.69	0.65	0.65	0.65
walking with load (3 kg for 70 m/min)	0.76	0.76	0.74	0.73
walking with load (5 kg for 55 m/min)	0.71	0.72	0.71	0.70
ascending stairs	1.20	1.16	1.14	1.14
jogging (130 m/min)	1.67	1.65	1.63	1.62
Total activities	0.66	0.66	0.66	0.65

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
