# Peer review of "Simple Prediction of Metabolic Equivalents of Daily Activities Using Heart Rate Monitor without Calibration of Individuals"

_ijerph, 2019, doi:10.3390/ijerph17010216_

Round 1

Reviewer 1 Report

The authors have addressed my comments. 

Author Response

We appreciate that the reviewers’ comments were highly insightful and enabled us to greatly improve the quality of our manuscript. We have changed our manuscript as below.

The authors have addressed my comments.

Thank you very much for your patience and helpful comments.

Reviewer 2 Report

Dear authors,

I want to congratulate to authors by the good work that they have realysed. I think that this paper must be published and I only have the next minor comments:

For a better impact of the paper in the database I recommend don´t repeat the term included in the title like keywords. It´s necessary to include the foundation of the METs methodology in the introduction. In the participants section authors must specify what it´s “physically active”. It´s necessary include if participant comply the recommendations of OMS or ACSM, for example.

Author Response

This manuscript is a resubmission of an earlier submission. The following is a list of the peer review reports and author responses from that submission.

Round 1

Reviewer 1 Report

The authors tested the regression capabilities of %HRR, age, fitness level, resting HR to assess MET.  They used multi-collinearity for feature selection.  %HRR was found to be the principal factor for MET estimation, which is in agreement with previous studies. Several models were tested, including different set of features.  The results show promising performance in the regression models.  The manuscript requires work to convey the concepts, measurements, methods, and results more clearly.  Furthermore, some other metrics should be used to evaluate the obtained models, and a more detailed analysis of the potentiality and limitations of the results should be provided.

Comments:

The introduction contains the scientific problem and details to understand the importance of the study. However, it need work to improve clarity.  The authors should explain straight forward the unmet need, and what is available in the field, and express clearly why that is not enough. Then be clear on why your approach is novel and innovative, and provides relevant information. Energy expenditure vs. metabolic equivalent. It seems like the authors tried to explain the relationship between the two in the introduction, but it wasn’t quite clear. Provide more details on the “complexity of the procedure” %HRR is not defined/explained in detail, in the manuscript. This and other concepts cannot be assumed known by the readers. “from each age group”. What groups? The groups were not defined.  Table 2 solves this question, but it should be explained in the text. Why was the body weight measured with a bioelectrical impedance analyzer? Why didn’t you use a scale for that? What type of HR measurement was performed (e.g. ECG, PPG, other)? R-peaks and R-R intervals are not defined. The manuscript requires a table with a list of measured parameters, and the method employed to collect such data. It is very confusing to read the protocol section. I couldn’t understand how MET was computed. You need to add an equation and explain it clearly.  Was MET measured or calculated? What device? What other values besides EE are necessary? “and only one of them should be used in the model”. Why can’t HR and %HRR be both included in the regression model? If they are only moderately correlated, both could provide useful information to the model. Other methods for feature selection should be used to discard irrelevant or redundant parameters.  How was correlation computed? Overall subjects mean or subject by subject? To make clear the validation approach, was it leave-one-out or leave-one-subject-out? Is that equivalent in your study? How were those 673 dots shown in figure 3 obtained? Are you holding out a condition/subject, but including all the other conditions for the subject in each model? How was the overall performance computed in your approach (e.g. average of errors for each leave-out run)? Authors should report RMSE (root mean square error) for the regression models. It is a sensitive index of regression performance. What is an acceptable range of error for MET? In other words, how bad is to have a MET estimation that is off by 18%? What is a possible consequence of that, if any? Figure 3 does not have values in the x-axis. “we recalculated MPE using the hold-out method. We divided all subjects into two parts in the ratio three to two for the model development group and the validation group, and calculated MPE of each activity. This was repeated 10,000 times and the largest values of MPE of each activity among the tests were used for comparison with previous studies.” Why were the largest MPE values selected for comparison with other studies? For a fair comparison, the average MPE of the 10,000 different times should be reported, because that is the fair measure of performance. All this details on hold-out or leave-out should be clearly explained in the methods section, not in the discussion section.

Other comments:

HR was defined in the abstract, but it must be defined the first time it’s used in the body of the manuscript. English grammar requires revision throughout the manuscript.

Reviewer 2 Report

The manuscript “Prediction of metabolic equivalent of daily activities using heart rate monitor and anthropometry measures” reports data from a study investigating the ability to predict METs for daily activities. Overall, the manuscript is well written, the study is well designed and the results are interesting. There are a few minor concerns.

In the Methods, Participants: the authors mention that 5-6 individuals were recruited for each “age group”. Please include the age groups in this section. Table 1 provides a list of daily activities that the participants completed for the study. Include the time spent doing the activities in the methods. “Cellular touching” should be changed, as it not descriptive of any activity.  “Moving load” should include the weight/size of load. Discussion: It is interesting that weight did not predict METs, but height did. Since the range of weights was small and the weight between participants were similar, could this be a potential reason for the lack of predictive value of weight? Would the authors predict that if there was a larger weight range, weight would be a predictor of MET?

Round 2

Reviewer 1 Report

The authors made changes that considerably improved the quality of the manuscript. The suitability of the feature selection strategy is still a matter of concern.

There is a plethora of feature-selection methods that you can use to avoid redundancy of the features. Correlation is not a suitable measure of redundancy.  I suggest reading this paper: http://www.jmlr.org/papers/v13/brown12a.html. mRMR is a good way to start (http://home.penglab.com/proj/mRMR/). Or you could use the Feature Selection Toolbox for C, Java and Matlab, available in http://www.cs.man.ac.uk/~pococka4/FEAST.html In fact, given the small amount of feature (4) and subjects (42), you could try all the possible combinations of features (instead of dealing with feature selection at all) and validate each combination using leave-one-subject-out and compare them using RMSE or any measure of performance you consider suitable for comparison with previous studies. Other studies have tried all possible combinations, even with larger amount of features (for classification purposes in the example provided, but can be used identically for regression), with a limited dataset like yours (https://doi.org/10.3390/bs9040045). RMSE values were added to the bottom of Table 6. However, there is no mention of it in the methods, results and discussion section.  It is not clear how they computed this value. RMSE is a stronger measure of error than MPE, and could be provided in the same fashion than MPE or MAPE. The only reason to report MPE is to compare to previous studies, if they only provided such measures.  Please comment on that. The manuscript says now that average MPE was reported now in table 6. However, the table was not changed at all. The previous version said that you were reporting the minimum MPE.  Please clarify.  Was also average reported for MAPE? Is the best (lowest) MAPE of 10,000 runs being reported? Comment on the last sentence of the introduction section: In my opinion you validated your model using cross-validation approach and measuring regression error, and evaluated it by comparing to other studies. Comparing to previous studies is not intended for validation of a model.